# Are teenagers in Kenya physically active? The nexus between physical activity and nutrition status of Kenyan teenagers: A cross-sectional study

**Moses Amram Kutwah** ⓘ*, **Dorcus Mbithe David-Kigaru, Joseph Kobia**

Department of Food, Nutrition and Dietetics, School of Health Sciences, Kenyatta University, Nairobi, Kenya

* kutwahmoses@gmail.com

## Abstract

Insufficient physical activity and poor nutrition during teenagerhood increase the risk of obesity, diabetes, and other chronic conditions. This study assessed physical activity levels and nutritional status of teenagers aged 13–19 years old in Machakos County, Kenya. A cross-sectional design was used to recruit 357 teenagers. Probability proportionate sampling and simple random methods were used to select households with teenagers. Socio-demographic data were collected by using an interviewer-administered questionnaire. Anthropometric measurements were taken by using a digital scale, height using a height board and waist circumference using a tape measure. The GPAQ-A was used to assess participants' physical activity levels. The WHO AnthroPlus v1.0.4 software was used to analyze anthropometric data. Waist circumference was calculated using the International Diabetes Federation (IDF) cut-off values. Bivariate and multivariate logistic regression analysis was performed to establish relationships between variables and control for potential confounding variables. Overall, 56.0% of teenagers were inactive, with females 56.5% reporting lower activity levels than males, 43.5% (p = 0.01). The overall prevalence of underweight and overweight based on BMI-for-age Z-scores was 8.1% and 9.2% respectively. Sex differences were noted; 10.1% of male teens were underweight, while 16.8% of female teens were overweight. Based on waist circumference, 3.4% of females were overweight, while no males were (p = 0.01). Being a male (Adjusted O.R. = 0.43, 95% CI: 0.18-0.99, p-value 0.04) and falling between ages 15–19 years old (Crude O.R. = 0.28, 95% CI: 0.13-0.62, p-value 0.00) were significantly associated with underweight. However, being female (Crude O.R. = 8.75, 95% CI: 3.06-25.43, p-value 0.00) and (Adjusted O.R. = 8.82, 95% CI: 2.98-25.94, p-value 0.00), teenager belonging to household with 3–6 members (Adjusted O.R. = 3.72, 95% CI: 2.95-14.63, p-value 0.04) and maternal education of secondary school (Crude O.R. = 0.31, 95% CI: 0.11-0.84, p-value 0.02) and (Adjusted O.R. = 0.29, 95% CI: 0.09-0.93,

**Data availability statement:** All relevant data are within the paper and its Supporting information files.

**Funding:** The author(s) received no specific funding for this work.

**Competing interests:** The authors have declared that no competing interests exist.

p-value 0.03) were significantly associated with overweight. A considerable number of Kenyan teenagers do not meet recommended physical activity levels, with implications for their nutritional status. Physical inactivity is clearly linked to poor nutritional outcomes.

## Introduction

Teenagerhood is a transitional stage in life marked by significant psychological, physical and emotional changes during which lifestyle habits are established, including physical activity and nutrition [1]. In Kenya, specifically in areas such as Machakos County, understanding the physical activity levels of teenagers and how they relate to their nutritional status is vital to aid in developing targeted nutritional interventions. Globally, there is a growing trend of physical inactivity among adolescents. The World Health Organization (WHO) records that in 2018, 81% of adolescents aged 11–17 years were physically inactive, with higher inactivity rates reported amongst girls compared to boys. This trend was ascribed to increased sedentary lifestyles, such as long hours of screen time, and decreased participation in physical education and recreational activities [1]. At the same time, the nutrition transition, a shift from traditional diets high in cereals and fiber to Western diets rich in sugars and fats, is attributed to the rise in overweight and obesity among adolescents, whilst undernutrition persists across population groups. This dual burden is a risk to overall well-being worldwide [2–4]. In Africa, the situation depicts a paradoxical interplay between undernutrition and evolving overnutrition [2].

Empirical evidence shows that, whereas undernutrition remains prevalent, there is a significant upsurge in overweight and obesity amongst adolescents, specifically in urban settings [5]. This change is partly attributed to rapid urbanization, which influences dietary patterns while at the same time lessens opportunities to engage in physical activity. For instance, in Ethiopia, 4.7% of adolescents were thin, 5.2% were stunted and 5.0% were overweight/obese, highlighting the existence of both undernutrition and overnutrition in the same population age group [6]. Simultaneously, the increase in sedentary lifestyles and consumption of energy-dense diets has contributed to rising malnutrition rates amongst African youth [7]. Kenya exemplifies this dual burden of malnutrition. A notable portion of Kenyan adolescents suffer from undernutrition, with stunting and micronutrient deficiencies being prevalent, especially in rural regions [8]. Conversely, urban areas have reportedly increased rates of overweight and obesity amongst teenagers, linked to lifestyle changes and dietary shifts [9]. National surveys have highlighted that 81% of Kenyan adults are insufficiently physically active, with men and women having 83% and 79% prevalence, respectively [10]. However, data specific to adolescents' physical activity levels are limited, underscoring the need for focused research in this population group. Yet it is known that changes in physical inactivity and poor nutritional status in this period can lead to health problems later in life, such as obesity and associated comorbidities.

Assessing physical activity levels and their connection to the nutritional status of teenagers in Machakos County is important for various reasons. First, teenagerhood is a

formative stage where lifelong health behaviors are established; therefore, interventions during this period have possible lasting impacts. Second, Machakos County, with its unique socio-economic and cultural dynamics, may pose distinct challenges and opportunities associated with teenage health that are less documented in broader national studies. Third, an understanding of the local context of physical activity levels and nutrition status among teenagers will inform the development of tailored public health strategies geared towards combating both undernutrition and the emerging trends of overnutrition. Such targeted interventions are imperative to improving the overall health outcomes of the teenage population in the area. Hence, this study aimed to assess the physical activity levels and their connection to the nutritional status of teenagers (13–19 years).

## Materials and methods

### Ethics statement

Ethical approval was granted by the Kenyatta University Ethical Review Committee (KUERC; PKU/2662/E1786), which reviewed the study protocol to safeguard participant rights and welfare. A research permit was granted by the National Commission for Science, Technology and Innovation (NACOSTI; NACOSTI/P22/22122), which oversees research licensing and institutional accreditation at the national level in Kenya. The County Director Medical Services and Research, Machakos County Government (MKS/DHES/RSCH/VOL1/326) granted permission to interview residents of Machakos County. Prior to enrollment, the study procedures were explained to all potential participants. Written informed consent was obtained from all participants aged 18 years and above. For participants under the age of 18, written informed consent was provided by a parent or a legal guardian, and the participants themselves provided written assent.

### Setting and research design

This study was conducted in Mulolongo, Mavoko Sub-County, Machakos County. Machakos County has an estimated overall population of 1,421,932. The youth constitute 29.2% while the elderly (65 and above years) constitutes 5.6% against the national tally of 29.0% and 3.9% respectively. In terms of population size, as of 2019, children aged between 0 and 14 years comprise 32.6% of the total population (N = 1,421,932), while those aged between 15 and 34 (youth) comprise 35.8% of the total population. Mavoko Sub-County was purposively selected since it had the highest population, 22.6% of the total population in the County. This population is attributed to the fact that it is an industrial town, fast developing in terms of infrastructure, real estate ventures and its proximity to Nairobi City. Mavoko/Athi River Sub-County has 4 administrative wards: Athi River, Kinanie, Muthwani Mulolongo wards. Mulolongo ward was purposively selected because of its setup. It is the smallest ward yet the most densely populated ward of Mavoko Sub-County; the majority of its rental/residential homes are alongside the Mombasa highway [11]. This work adopted a cross-sectional analytical study design to achieve the objective. This approach was appropriate to (a) quantify the current burden of malnutrition, and (b) provide an empirical foundation for understanding the interrelationships between key nutrition indicators, using BMI for age z scores as a primary indicator for nutritional status of population under study.

### Study population and sampling procedure

**Study population.** The primary study population was teenagers, both males and females, 13–19 years old in Mulolongo, Mavoko Sub-County.

**Determination of sample size.** The target sample size was calculated based on the Cochran Formula (1997): n (pq) $Z^2/d^2$. Where n = the least sample size in a population with more than 10,000 people, Z = 99% confidence level for the normal deviation linked with the level of significance set at 2.576, and p = population proportion of overweight among teenagers aged 13–18 years in Nairobi County, 13.7% [12]. q = proportion without the characteristics being measured (1-p), d = the degree of accuracy required, generally, 0.05. Therefore, when the values were substituted to the formulae; n = 0.137 x (1- 0.137) x $(2.576)^2 \div (0.05)^2 = 313.8 = 314$ teenagers. Adding 10% [13] for non-response, the calculated sample size was 345 teenagers.

**Sampling procedure.** Mulolongo ward was purposively selected since it is the most densely populated ward in Mavoko Sub-County. A proportionate sampling technique based on population density was used, given the ward was administratively divided into 4 community health units/villages: Mulolongo A/Phase 1 and 2, Mulolongo B/Phase 3, Syokimau and Sabaki. A simple random sampling technique was used to select households with teenagers aged 13–19 years who participated in the study. Once the enumerators reached the midpoint of the selected village 'spin bottle method' was used to identify the first household with a teenager to interview. In this method, the enumerators identified a flat surface and spun the bottle until it took the direction facing the mouth of the bottle. The first teenager in the household in that direction was selected, verbal and written assent and consent where necessary were first sought and then the team went ahead to administer the questionnaire and took anthropometric measurements of the teenager. Once the first teenager was fully assessed, the enumerators repeated the spin bottle method outside of the already sampled household to get the direction of the next household. An enumerator who reached the end of the village/estate before completing the numbers required walked back to the centre of the village and spun the bottle again. In case the enumerator double-selected a previously sampled household, that household was excluded and the exercise was repeated until all eligible households with teenagers were sampled.

## Methods of data collection

**Recruitment period for this study.** Data was collected at households between the 4th of May 2023 and the 27th of August 2023 (See S1 Dataset).

**Structured questionnaire.** The questionnaire had three sections: socio-demographic characteristics of the teenagers and household, anthropometric assessment and the physical activity sections (S1 Table). The socio-demographic data collected included participants' age, gender, household size, educational attainment and religion. Anthropometric measurements taken included weight, height and waist circumference of the participants. The physical activity section captured participants' engagement in general physical activity, like during physical education (PE) classes in school, lunch hours, right after school, recreational activities in the evenings and also weekends while at home. The nutritional status of the participants was evaluated through the collected anthropometric measures. Weight and height were used to compute BMI-for-age z-scores, while abdominal adiposity was assessed by applying standard cut-off points to waist circumference measurements.

**Pre-testing of the questionnaire, validity and reliability.** The questionnaire was pre-tested to check for clarity of content, adequacy, and flow of the questions. The questionnaire was further validated by research supervisors, the Food, Nutrition, and Dietetics Department and a group of nutrition and dietetics experts by reviewing the content, refining questions, and ensuring methodological soundness to confirm relevance and validity as per the objective of the study. The use of validated and standard tools like the Global Physical Activity Questionnaire for Adolescents (GPAQ-A) ensured validity. To check for reliability, the test-retest technique was performed by administering the same questionnaire in more than one household in the Athi River ward before actual data collection to establish stability over time.

**Questionnaire administration.** The research employed a structured questionnaire, administered through face-to-face interviews. The data collection protocol involved several steps to ensure ethical compliance; (a) all participants were clearly informed on the study's purpose and procedures, (b) written and verbal informed consent was sought from participants aged 18 and above and (c) for participants aged below 18, written informed consent was provided by a parent or legal guardian, and the minor provided written assent before administration of the questionnaire. The teenagers were allowed to respond freely and in their language of choice. To ensure participant confidentiality and privacy, all data sets were anonymized.

**Engagement in physical activity levels.** Data on engagement in physical activity levels and practices among the teenagers were collected using the Global Physical Activity Questionnaire for Adolescents (GPAQ-A), developed by WHO [14]. The questionnaire assessed the frequency and intensity of physical activity among the teenagers and captured the

 

various types of activities: sports, recreational exercise, active transportation, school-based physical activity and also tracked teenagers' sedentary behaviours.

**Anthropometry (weight, height and waist circumference (WC) measurements).** Height measurements were taken using a wooden adult height board (UNICEF), which was mounted on a wall and participants were asked to stand straight with their back against the height board. The study participants were requested to remove shoes, undo hairstyle gears, eyes and ears lined horizontally, inhaling and thereafter step on the height board. For accuracy purposes, the height readings were measured twice for each teenager and thereafter, the average was calculated and recorded. The height calibrations were recorded to the nearest 0.1 cm. Weight was measured using the Omron HN289 personal weighing scale, which was calibrated at the beginning of each weight-taking session. The teenager was requested to remove weighty clothes and stand in a firm position with their arms along their body, and stand on the scale barefoot. Similarly, the 1st and 2nd readings of weight were measured and recorded to the nearest 0.1 kg on the anthropometric data sheet then the average was calculated to get the accurate weight of the teenager.

Waist circumference (WC) was measured and recorded to the nearest 0.1 cm using an inelastic tape (UNICEF). Participants were asked to stand up straight and locate the middle point (belly button) between the bottom of the ribs and the top of the hips. Upon breathing out gently, the tape measure was placed on the skin and measurements were taken after the participant exhaled. This was repeated twice for purposes of accuracy and measurements were recorded to the nearest 0.1 cm.

**Digital data capture.** The KoBo platform was used to collect data in the form of a digital version of the questionnaire. The pre-designed version of the form was uploaded and stored in a remote server of the Kobo Toolbox. Once on the server, the form was deployed and validated for use. KoBo Collect app, accessible from App-store or Google Play Store, was installed on all the Android phones of the research team. An Android version of the digitized questionnaire was then installed on the smartphones for all research team. The research team was then inducted on the use of the digital questionnaire in Android format; installing, navigating through the form, collecting, saving and submitting the filled forms to the server. The host server was protected using password-encrypted logs. Data collected was submitted to the remote server and aggregated into one sheet that was later transferred to an Excel spreadsheet for data sorting, cleaning, and management. Only authorized personnel (principal investigators) were allowed to access the data sets.

## Data analysis plan

Physical activity level of teenagers was analyzed and classified into the 3 scoring methods for physical activity levels as per the GPAQ-A guidelines (1 = low, 2 = moderate and 3 = high). The tool captured engagement in general physical activity, like during physical education (PE) classes in school, lunch hours, right after school, recreational activities in the evenings and also weekends while at home. The mean of all the activities as selected between 1–5 for each of the six physical activity domains, was computed to get the summary score for each teenager per week (7-day period). Physical activity classification using the GPAQ-A scoring system allowed for consistent categorization of activity levels across low, moderate, and high intensity, enabling comparative analysis. Further, the WHO AnthroPlus software for the global application of the WHO Reference 2007 for 5–19 years was useful in analyzing anthropometric data for the teenagers. The nutrition status of the teenagers was determined with reference to the WHO growth standards of BMI-for-age Z-scores and waist circumference cut-off values for the different genders. BMI-for-age Z-scores cut off points of < −3.0, < −2.0, < −1.0, > 1.0, > 2.0, > 3.0 are recommended to define severely underweight, moderately underweight, underweight, at risk of overweight, overweight and obese [15]. Waist circumference cut-offs, based on gender, were calculated using the International Diabetes Federation (IDF) cut-off values. According to these values, males with a waist circumference of ≥90 cm and females with a waist circumference of ≥80 cm are considered at risk for abdominal obesity and metabolic abnormalities (Ozturk et al., 2015 [16]; Yamanaka et al., 2021 [17]). The use of BMI-for-age z-scores and waist circumference cut-offs provided standardized, age-appropriate indicators of nutritional status, aligning with WHO growth reference guidelines. Binary

PLOS Global Public Health

logistic regression was an appropriate choice for examining the likelihood of categorical outcomes, precisely underweight and overweight, based on predictor variables such as household characteristics and socio-demographic factors. First, Crude Odds Ratios (COR) based on 95% confidence intervals were generated from a simple model with only one variable (Model I) at a time. Secondly, Adjusted Odds Ratios (AOR) for a model with all the variables (Model II) of interest in a pre-defined domain by the researcher to unveil how the modification affects the impact of a certain explanatory variable were then computed in the binary logistic regression analysis model based on 95% confidence intervals. The chronological use of Crude Odds Ratios (COR) and Adjusted Odds Ratios (AOR) enabled the researchers to first identify univariate associations and then control for potential confounders in multivariate models, thereby enhancing the validity and interpretability of the findings. This methodological approach safeguarded rigorous analysis of multifaceted relationships between variables within the study population. This study adhered to the STROBE-nut reporting guidelines for observational nutritional epidemiology research [18].

## Results

### Response rate

The final sample size obtained was 357 teenagers. This surpassed the minimum sample size required for the study.

### Demographic characteristics of the teenagers

Demographic data were collected from both the teenagers and the parents/guardians of the household where the teenagers resided.

**Characteristics of the study teenagers.** There was no significant difference between the number of respondents, males 49.9% and females 50.1%, p-value > 0.05. The mean (SD) age of the teenagers was 16.22 ± 1.51. A large proportion of the teenagers sampled were 15–19 years old, 79.0%. About 72.5% of the study participants had secondary education, with no significant difference between males and females. The most commonly practiced religion was Christianity, 96.4%. The religion of teenagers was the same as that of their parents/guardians (Table 1).

**Characteristics of the teenagers' households.** In this study, most of the households consisted of 3–6 members, 70.3%. In the majority of the households, the teenagers were living with their parents; mothers 35.7%, fathers 28.3% and their siblings 28.4% during the time of the study. In this study, 40.3% (p-value 0.027) and 35.3% of the teenagers were not sure of the education level of their fathers and mothers, respectively, during the time of the survey (Table 2).

### Physical activity levels of the teenagers

Findings indicated that teenagers are not active. Over the past week of data collection, the majority of teenagers, specifically 56.0%, were not active. Notably, females were more affected, with 56.5% reporting insufficient activity compared to 43.5% of males. This difference was statistically significant, p-value 0.01. In terms of physical education, 35.9% of teenagers occasionally participated in these classes, which also showed a significant difference between males and females, p-value 0.01. During lunch breaks, a substantial portion of teenagers, 76.2% remained inactive, spending their time sitting while talking, reading, or doing schoolwork, again with a significant difference among males and females, p-value 0.04. Post-school and evening hours also reflected low activity levels, with 51.8% and 37.0% of respondents, respectively, engaging in no physical activity, p-value 0.05. This pattern was consistent over the weekends, where 40.6% of teenagers similarly did not engage in physical activity. Overall, for the past seven days, the majority of respondents engaged in minimal physical activity, where 42.3% reportedly engaging in low physical activity levels. This trend had a statistically significant difference, underscoring a need for increased awareness and intervention to promote physical activity among teenagers in this area. There was a significant difference in physical activity levels between male and female teenagers throughout the 7 days, p-value 0.00. Specifically, female teenagers were less active compared to their male counterparts.

**Table 1. Distribution of teenagers by demographic characteristics based on sex.**

| Characteristics of the teenagers (N = 357) | | Gender | | | χ² p-value |
|---|---|---|---|---|---|
| | Total (N = 357) | Male (n = 178) | Female (n = 179) | | |
| | n(%) | n(%) | n(%) | | |
| **Age clusters of teenagers\* (mean age; 16.22 ± 1.51)** | | | | | |
| 13-14 years | 75(21.0) | 34(45.3) | 41(54.7) | | 0.378 |
| 15-19 years | 282(79.0) | 144(51.1) | 138(48.9) | | |
| **Highest level of education** | | | | | |
| Secondary | 259(72.5) | 129(49.8) | 130(50.2) | | 0.819 |
| Primary | 66(18.5) | 31(47.0) | 35(53.0) | | |
| University | 21(5.9) | 13(61.9) | 8(38.1) | | |
| Vocational/technical training | 9(2.5) | 4(44.4) | 5(55.6) | | |
| Not finalized education | 2(0.6) | 1(50.0) | 1(50.0) | | |
| **Current religion** | | | | | |
| Christian | 344(96.4) | 169(49.1) | 175(50.9) | | 0.155 |
| Muslim | 13(3.6) | 9(69.2) | 4(30.8) | | |

**Table legend:** Descriptive statistics were generated to summarize variable proportions. Cross-tabulations were employed to assess the relationship between gender and demographic characteristics of the teenagers. Gender-based differences were evaluated using Pearson's chi-square (χ²) test, with statistical significance set at $p < 0.05$. \*WHO classification age.

This indicated a notable disparity in physical activity engagement, highlighting the need for targeted interventions to address and improve physical activity among female teenagers in Mulolongo Ward (Table 3).

### Nutrition status of teenagers 13–19 years in Mulolongo Ward

**Teenagers' nutrition status by BMI for age Z-scores and age.** The study found that the overall prevalence of severe underweight, moderate underweight, at-risk of overweight, and overweight based on BMI-for-age Z-scores was 1.4%, 6.7%, 9.0%, and 0.3%, respectively. When stratified by age, among respondents aged 13–14 years, the prevalence of severe underweight was 4.0%, moderate underweight 13.3%, at-risk of overweight 9.3%, and no cases of overweight were recorded. Conversely, among respondents aged 15–19 years, the prevalence rates were 0.7% for severe underweight, 5.0% for moderate underweight, 10.6% for at-risk of overweight, and 0.4% for overweight. Notably, 82.6% of all participants had a normal nutritional status. Within age groups, 73.4% of 13–14-year-olds and 83.3% of 15–19-year-olds were classified as having normal nutritional status (Table 4).

**Teenagers' nutrition status based on BMI for age Z-scores by sex and age.** The nutritional status of respondents was analyzed by sex. Among male respondents, the prevalence of moderate underweight was higher (8.4%) than severe underweight (1.7%) and being at risk of overweight (1.2%). Overall, 88.7% of males had normal nutritional status based on BMI-for-age Z-scores. Within the male age groups, 13–14-year-olds showed a higher prevalence of moderate underweight (17.6%) compared to those aged 15–19 years (6.3%).

Among female respondents, the prevalence of being at risk of overweight was notably higher (16.2%) compared to moderate underweight (5.0%), severe underweight (1.1%), and overweight (0.6%). A total of 77.1% of females had a normal nutritional status. Age-specific analysis revealed that females aged 15–19 years were more at risk of being overweight (16.7%) compared to those aged 13–14 years (14.6%). However, moderate underweight was more prevalent among females aged 13–14 years (9.8%) than those aged 15–19 years (3.6%) (Table 5).

**Table 2. Distribution of teenagers by household demographics and socio-economic characteristics based on sex.**

| Characteristics of the household (N=357) | | Gender | | χ² p-value |
|---|---|---|---|---|
| | Total (N=357) | Male (n=178) | Female (n=179) | |
| | n(%) | n(%) | n(%) | |
| **Number of people in the household (Reference period is the last 7 days)** | | | | |
| 3 to 6 members | 251(70.3) | 121(48.2) | 130(51.8) | 0.325 |
| 1 to 3 members | 73(20.4) | 42(57.5) | 31(42.5) | |
| 7 to 9 members | 33(9.3) | 15(45.5) | 18(54.5) | |
| **Person living with (multiple response N=856)** | | | | |
| Mother | 306(35.7) | 145(47.4) | 161(52.6) | – |
| Siblings | 243(28.4) | 118(48.6) | 125(51.4) | |
| Father | 242(28.3) | 131(54.1) | 111(45.9) | |
| Other relatives | 35(4.1) | 17(48.6) | 18(51.4) | |
| Others, please specify | 23(2.7) | 12(52.2) | 11(47.8) | |
| Self, alone | 7(0.8) | 6(85.7) | 1(14.3) | |
| **Father's/guardian's highest level of education** | | | | |
| I don't know/not sure | 144(40.3) | 72(50.0) | 72(50.0) | 0.027* |
| Secondary | 80(22.4) | 44(55.0) | 36(45.0) | |
| University | 55(15.4) | 31(56.4) | 24(43.6) | |
| Vocational/technical training | 31(8.7) | 7(22.6) | 24(77.4) | |
| Others (Please specify) | 22(6.2) | 9(40.9) | 13(59.1) | |
| Not finalized education | 1(0.3) | 0(0.0) | 1(100.0) | |
| Primary | 24(6.7) | 15(62.5) | 9(37.5) | |
| **Mother's/guardian's highest level of education** | | | | |
| I don't know/not sure | 126(35.3) | 64(50.8) | 62(49.2) | 0.745 |
| Secondary | 108(30.3) | 51(47.2) | 57(52.8) | |
| Primary | 49(13.7) | 27(55.1) | 22(44.9) | |
| University | 39(10.9) | 20(51.3) | 19(48.7) | |
| Vocational/technical training | 31(8.7) | 13(41.9) | 18(58.1) | |
| Others (Please specify) | 4(1.1) | 3(75.0) | 1(25.0) | |

**Table legend:** Descriptive statistics were generated to summarize variable proportions. Cross-tabulations were employed to assess the relationship between gender and the household's demographics and socio-economic characteristics. Gender-based differences were evaluated using Pearson's chi-square (χ²) test, with statistical significance set at p<0.05.*Significant difference between males and females (Pearson chi-square, χ2 test and p-value <0.05). Education was asked for every teenager, irrespective of whether the teenager is living with their father/mother or not, during the time of the survey.

**Waist circumference of respondents by age and sex.** Waist circumference, an anthropometric indicator of abdominal obesity, serves as an independent predictor of insulin resistance and other comorbidities [19,20]. The classification of individuals at risk was based on the International Diabetes Federation (IDF) cutoff values, which define males with a waist circumference of ≥90 cm and females with a waist circumference of ≥80 cm as at risk for metabolic abnormalities. These abnormalities include impaired glucose tolerance, reduced insulin sensitivity, and adverse lipid profiles, all of which are recognized risk factors for type 2 diabetes and cardiovascular disease (CVD) [21,22]. Analysis revealed that 3.4% of females were classified as overweight, whereas no males fell into this category. This gender-based disparity was statistically significant, p=0.01. Furthermore, within age groups, 3.6% of females aged 15–19 years were overweight compared to 2.4% of females aged 13–14 years, a difference that was also statistically significant, p=0.02.

**Table 3. Distribution of respondents by sex on engagement in physical activity level over 7 days.**

| Physical activity level of teenagers in the last 7 days (N=357) | | Gender | | χ² p-value |
|---|---|---|---|---|
| | Total | Male | Female | |
| | n(%) | n(%) | n(%) | |
| **Physical activity during physical education (PE) classes lasts 7 days.** | | | | |
| I don't to PE | 106(29.7) | 50(47.2) | 56(52.8) | 0.01* |
| Hardly ever | 33(9.2) | 13(39.4) | 20(60.6) | |
| Sometimes | 128(35.9) | 56(43.7) | 72(56.3) | |
| Quite often | 51(14.3) | 35(68.6) | 16(31.4) | |
| Always | 39(10.9) | 24(61.5) | 15(38.5) | |
| **PA during lunch (besides eating lunch), last 7 days** | | | | |
| Sat down (talking, reading, doing schoolwork) | 272(76.2) | 129(47.4) | 143(52.6) | 0.04* |
| Stood around or walked around | 63(17.6) | 38(60.3) | 25(39.7) | |
| Ran or played a little bit | 7(2.0) | 1(14.3) | 6(85.7) | |
| Ran around and played quite a bit | 9(2.5) | 5(55.6) | 4(44.4) | |
| Ran and played hard most of the time | 6(1.7) | 5(83.3) | 1(16.7) | |
| **PA right after school, last 7 days** | | | | |
| None | 185(51.8) | 94(50.8) | 91(49.2) | 0.14 |
| 1-time last week | 46(12.9) | 17(37.0) | 29(63.0) | |
| 2- or 3-times last week | 78(21.8) | 37(47.4) | 41(52.6) | |
| 4 times last week | 22(6.2) | 15(68.2) | 7(31.8) | |
| 5 times last week | 26(7.3) | 15(57.7) | 11(42.3) | |
| **PA on evenings, last 7 days** | | | | |
| None | 132(37.0) | 57(43.2) | 75(56.8) | 0.05* |
| 1-time last week | 49(13.7) | 22(44.9) | 27(55.1) | |
| 2- or 3-times last week | 110(30.8) | 56(50.9) | 54(49.1) | |
| 4- or 5-times last week | 41(11.5) | 26(63.4) | 15(36.6) | |
| 6- or 7-times last week | 25(7.0) | 17(68.0) | 8(32.0) | |
| **No. of times engaged in PA last weekend,** | | | | |
| None | 145(40.6) | 64(44.1) | 81(55.9) | 0.07 |
| 1 time | 87(24.4) | 43(49.4) | 44(50.6) | |
| 2 – 3 times | 106(29.7) | 60(56.6) | 46(43.4) | |
| 4 – 5 times | 9(2.5) | 3(33.3) | 6(66.7) | |
| 6 or more times | 10(2.8) | 8(80.0) | 2(20.0) | |
| **Summary of PA, the last 7 days** | | | | |
| Little physical effort | 151(42.4) | 60(39.7) | 91(60.3) | 0.00* |
| (1–2 times last week) | 109(30.5) | 48(44.0) | 61(56.0) | |
| (3–4 times last week) | 60(16.8) | 42(70.0) | 18(30.0) | |
| (5–6 times last week) | 23(6.4) | 17(73.9) | 6(26.1) | |
| (7 or more times last week) | 14(3.9) | 11(78.6) | 3(21.4) | |
| **Combined physical activity for the last 7 days** | | | | |
| Low | 200(56.0) | 87(43.5) | 113(56.5) | 0.01* |
| Moderate | 152(42.6) | 87(57.2) | 65(42.8) | |
| High | 5(1.4) | 4(80.0) | 1(20.0) | |

**Table legend:** Descriptive statistics were generated to summarize variable proportions. Cross-tabulations were employed to assess the relationship between gender and engagement in physical activity levels over 7 days. Gender-based differences were evaluated using Pearson's chi-square ($\chi^2$) test, with statistical significance set at $p < 0.05$.*Significant difference between males and females (Pearson chi-square, $\chi^2$ test and p-value <0.05).

**Table 4. Distribution of respondents' nutritional status (BMI-for-age) by age.**

| Age groups | | BMI-for-age Z scores (BAZ) | | | | | | |
|---|---|---|---|---|---|---|---|---|
| Years | N | Severely underweight (<-3 z-scores) | Moderate underweight (<-2 z-scores) | Normal <-1 to <+1 z-score | Risk of overweight (>+1 z-score) | Overweight (>+2 z scores) | Mean | SD |
| | | n(%) | n(%) | n(%) | n(%) | n(%) | | |
| 13-14 | 75 | 3(4.0) | 10(13.3) | 55(73.4) | 7(9.3) | 0(0.0) | -0.5 | 1.18 |
| 15-19 | 282 | 2(0.7) | 14(5.0) | 235(83.3) | 30(10.6) | 1(0.4) | -0.24 | 1.0 |
| Total (15–1913 –19 ) | 357 | 5(1.4) | 24(6.7) | 295(82.6) | 32(9.0) | 1(0.3) | -0.97 | 1.53 |

**Table legend:** Classification of nutritional status of the teenagers based on the WHO BMI-for-age Z scores (BAZ) cut-off points.

**Source:** WHO BMI for age z-scores; https://www.who.int/tools/growth-reference-data-for-5to19-years/indicators/bmi-for-age.

**Table 5. Distribution of respondents' nutritional status (BMI-for-age) by sex and age.**

| Age groups | | BMI-for-age Z scores (BAZ) | | | | | | |
|---|---|---|---|---|---|---|---|---|
| Years | N | Severely underweight (<-3 z-scores) | Moderate underweight (<-2 z-scores) | Normal <-1 to <+1 z-score | Risk of overweight (>+1 z-score) | Overweight (>+2 z scores) | Mean | SD |
| | | n(%) | n(%) | n(%) | n(%) | n(%) | | |
| **Males** | | | | | | | | |
| 13-14 | 34 | 2(5.9) | 6(17.6) | 25(73.5) | 1(3.0) | 0(0.0) | -0.82 | 1.17 |
| 15-19 | 144 | 1(0.7) | 9(6.3) | 131(90.9) | 3(2.1) | 0(0.0) | -0.59 | 0.91 |
| Total (15–1913 –19 ) | 178 | 3(1.7) | 15(8.4) | 158(88.7) | 2(1.2) | 0(0.0) | 1.31 | 1.55 |
| **Females** | | | | | | | | |
| 13-14 | 41 | 1(2.4) | 4(9.8) | 30(73.2) | 6(14.6) | 0(0.0) | -0.24 | 1.14 |
| 15-19 | 138 | 1(0.7) | 5(3.6) | 108(78.3) | 23(16.7) | 1(0.7) | 0.13 | 0.96 |
| Total (15–1913 –19 ) | 179 | 2(1.1) | 9(5.0) | 138(77.1) | 29(16.2) | 1(0.6) | 0.78 | 1.01 |

**Table legend:** Classification of nutritional status of the teenagers based on the WHO BMI-for-age Z scores (BAZ) cut-off points by sex and age. None of the male teenagers was overweight or obese.

**Source:** WHO BMI for age z-scores; https://www.who.int/tools/growth-reference-data-for-5to19-years/indicators/bmi-for-age.

These findings suggest that females are at a higher risk of abdominal obesity and its associated comorbidities compared to males (Table 6).

### Relationship between demographics, socio-economic characteristics and underweight

Bivariate (Model I) and multivariate (Model II) logistic regression analyses showed significant relationships between characteristics of the teenager (gender and age cluster) and underweight. In the adjusted multivariate analysis (Model II), sex was significantly associated with underweight after controlling for potential confounding variables. Female teenagers were less likely to be underweight as compared to male teenagers at an (Adjusted O.R. = 0.43, 95% CI: 0.18-0.99, p-value 0.04), suggesting a notable sex-based disparity in nutritional status. In the unadjusted bivariate analysis (Model I), age cluster was significantly associated with underweight. Specifically, teenagers aged 15–19 years had lower odds of being underweight as compared to teenagers aged between 13–14 years (Crude O.R. = 0.28, 95% CI: 0.13-0.62, p-value 0.00). This implied that older teenagers were approximately 72% less likely to be underweight than their younger counterparts, indicating a protective effect of increasing age within this range. There were no significant relationships between the characteristics of the households where the respondents resided and underweight (Table 7).

**Table 6. Distribution of respondents' waist circumference by age and sex.**

| Age cluster and sex | Total | Normal | Overweight | χ² |
|---|---|---|---|---|
| | n(%) | n(%) | n(%) | p-value |
| **13-14 years** | | | | |
| Male | 34(45.3) | 34(100.0) | 0(0.0) | 0.36 |
| Female | 41(54.7) | 40(97.6) | 1(2.4) | |
| **15-19 years** | | | | |
| Male | 144(51.1) | 144(100.0) | 0(0.0) | 0.02* |
| Female | 138(48.9) | 133(96.4) | 5(3.6) | |
| **Total** | | | | |
| Male | 178(49.9) | 178(100.0) | 0(0.0) | 0.01* |
| Female | 179(50.1) | 173(96.6) | 6(3.4) | |

**Table legend:** Classification of nutritional status of the teenagers based on waist circumference cut-off points by sex and age cluster. Cross-tabulations were employed to assess the relationship between gender and age cluster and waist circumference classifications. Gender-based differences were evaluated using Pearson's chi-square (χ²) test, with statistical significance set at p < 0.05.*Significant difference between males and females (Pearson chi-square, χ2 test and p-value <0.05).

**Source**: The International Diabetes Federation criteria for ethnic or country-specific values for waist circumference (Males ≥ 90 cm, Females ≥ 80 cm).

## Relationship between demographics, socio-economic characteristics and overweight

Bivariate (Model I) and multivariate (Model II) logistic regression analyses were run to identify associations between the teenagers' individual characteristics, their household characteristics, and the likelihood of being overweight. In both the unadjusted bivariate (Model I) and adjusted multivariate (Model II) logistic regression analysis, sex was significantly associated with being overweight. Specifically, female respondents were more likely to be overweight compared to males at (Crude O.R. = 8.75, 95% CI: 3.06-25.43, p-value 0.00) while the (Adjusted O.R. = 8.82, 95% CI: 2.98-25.94, p-value 0.00) remained significantly higher, signifying a robust and statistically significant association after controlling for potential confounders. Further, in the adjusted multivariate analysis (Model II), household size was significantly associated with being overweight. Specifically, an increment of a teenager belonging to a household with 3–6 members had markedly higher odds of being overweight as compared to those who belonged to smaller households with 1–3 members. The (Adjusted O.R. = 3.72, 95% CI: 2.95-14.63, p-value 0.04) signified a statistically significant relationship after controlling for potential confounders.

Maternal education status showed a statistically significant association with a teenager's being overweight. Specifically, teenagers whose mothers had attained secondary education exhibited a greater likelihood of being overweight compared to those whose mothers had only primary education. This relationship was evident in both the unadjusted bivariate (Model I) (Crude O.R. = 0.31, 95% CI: 0.11-0.84, p-value 0.02) and adjusted multivariate (Model II) analyses (Adjusted O.R. = 0.29, 95% CI: 0.09-0.93, p-value 0.03), respectively, after controlling for other covariates. Notably, teenagers who reported not knowing their mother's education demonstrated a significant relationship with overweight in the bivariate analysis (Model I) (Crude O.R. = 0.22, 95% CI: 0.08-0.64, p-value 0.00). This implied that a lack of maternal education information was linked to an increased likelihood of being overweight (Table 8).

## Discussions

This study assessed physical activity (PA) and nutritional status of teenagers in Machakos County, Kenya. The study established that overall half 56.0% of the teenagers were physically inactive, with females being more inactive, 56.5%

**Table 7. Crude and adjusted odds ratios from bivariate (Model I) and multivariate (Model II) logistic regression analyses examining associations between demographics, socio-economic characteristics and underweight.**

| Characteristics of the teenager | Underweight | | Model I | | Model II | |
|---|---|---|---|---|---|---|
| | No | Yes | Sig | COR (95%CI) | Sig | AOR (95%CI) |
| | n(%) | n(%) | | | | |
| **Gender** | | | | | | |
| Male | 159(89.3) | 19(10.7) | 1 | | 1 | |
| Female | 169(94.4) | 10(5.6) | 0.08 | 0.49(0.22-1.09) | 0.04* | 0.43(0.18-0.99) |
| **Age cluster of the teenagers** | | | | | | |
| 13-14 years | 62(82.7) | 13(17.3) | 1 | | 1 | |
| 15-19 years | 266(94.3) | 16(5.7) | 0.00* | 0.28(0.13-0.62) | 0.36 | 0.57(0.17-1.93) |
| **Teenagers' highest level of education** | | | | | | |
| Not finalized | 2(100.0) | 0(0.0) | 1 | | 1 | |
| Primary | 54(81.8) | 12(18.2) | 0.99 | 359.40(0.00) | 0.99 | 403.00(0.00) |
| Secondary | 244(94.2) | 15(5.8) | 0.99 | 993.90(0.00) | 0.99 | 189.60(0.00) |
| University | 19(90.5) | 2(9.5) | 0.99 | 170.20(0.00) | 0.99 | 572.50(0.00) |
| Vocational/technical training | 9(100.0) | 0(0.0) | 1.00 | 1.00(0.00) | 1.00 | 10.00(0.00) |
| **Characteristics of the household** | | | | | | |
| **No. of people in household** | | | | | | |
| 1 to 3 members | 69(94.5) | 4(5.5) | 1 | | 1 | |
| 3 to 6 members | 227(90.4) | 24(9.6) | 0.28 | 1.82(0.61-5.43) | 0.18 | 2.48(0.64-9.58) |
| 7 to 9 members | 32(97.0) | 1(3.0) | 0.58 | 0.53(0.05-5.01) | 0.92 | 0.88(0.07-10.26) |
| **Person responsible for financing the food budget** | | | | | | |
| Father | 181(91.4) | 17(8.6) | 1 | | 1 | |
| Mother | 89(89.0) | 11(11.0) | 0.50 | 1.31(0.59-2.92) | 0.29 | 0.56(0.19-1.65) |
| Grandparents | 3(100.0) | 0(0.0) | 0.99 | 0.00(0.00) | 0.99 | 0.00(0.00) |
| Relatives (uncle, aunt, cousin) | 9(100.0) | 0(0.0) | 0.99 | 0.00(0.00) | 0.99 | 0.00(0.00) |
| Self | 5(83.3) | 1(16.7) | 0.50 | 2.12(0.23-19.29) | 0.49 | 2.56(0.17-37.95) |
| Others | 38(100.0) | 0(0.0) | 0.99 | 0.00(0.00) | 0.99 | 0.00(0.00) |
| I don't know/not sure | 3(100.0) | 0(0.0) | 0.99 | 0.00(0.00) | 0.99 | 0.00(0.00) |
| **Father's highest level of education** | | | | | | |
| Not finalized | 1(100.0) | 0(0.0) | 1 | | 1 | |
| Primary | 23(95.8) | 1(4.2) | 1.00 | 702.17(0.00) | 1.00 | 389.41(0.00) |
| Secondary | 75(93.8) | 5(6.3) | 1.00 | 107.60(0.00) | 1.00 | 953.23(0.00) |
| University | 54(98.2) | 1(1.8) | 1.00 | 299.00(0.00) | 1.00 | 888.52(0.00) |
| Vocational/technical training | 30(96.8) | 1(3.2) | 1.00 | 538.80(0.00) | 1.00 | 168.40(0.00) |
| Others | 21(95.5) | 1(4.5) | 1.00 | 769.00(0.00) | 1.00 | 546.54(0.00) |
| I don't know/not sure | 124(86.1) | 20(13.9) | 1.00 | 260.10(0.00) | 1.00 | 369.40(0.00) |
| **Mother's highest level of education** | | | | | | |
| Primary | 44(89.8) | 5(10.2) | 1 | | 1 | |
| Secondary | 99(91.7) | 9(8.3) | 0.70 | 0.80(0.25-2.52) | 0.76 | 0.81(0.21-3.06) |
| University | 38(97.4.) | 1(2.6) | 0.19 | 0.23(0.02-2.07) | 0.34 | 0.30(0.02-3.51) |
| Vocational/technical training | 31(100.0) | 0(0.0) | 0.99 | 0(0.00) | 0.99 | 0.00(0.00) |
| Others | 4(100.0) | 0(0.0) | 0.99 | 0(0.00) | 0.99 | 0.00(0.00) |
| I don't know/not sure | 112(88.9) | 14(11.1) | 0.86 | 1.10(.037-3.23) | 0.33 | 0.52(0.14-1.95) |
| **The household's main source of income** | | | | | | |
| Agriculture | 4(100.0) | 0(0.0) | 1 | | 1 | |
| Casual labour | 53(81.5) | 12(18.5) | 0.99 | 365.40(0.00) | 0.99 | 525.20(0.00) |

*(Continued)*

**Table 7.** (Continued)

| Characteristics of the teenager | Underweight | | Model I | | Model II | |
|---|---|---|---|---|---|---|
| | No | Yes | Sig | COR (95%CI) | Sig | AOR (95%CI) |
| | n(%) | n(%) | | | | |
| Self-employed running a business | 63(92.6) | 5(7.4) | 0.99 | 128.30(0.00) | 0.99 | 135.60(0.00) |
| Salaried employment | 194(95.1) | 10(4.9) | 0.99 | 832.97(0.00) | 0.99 | 990.87(0.00) |
| Petty trade | 5(83.3) | 1(16.7) | 0.99 | 323.60(0.00) | 0.99 | 628.10(0.00) |
| Others | 1(100.0) | 0(0.0) | 1.00 | 1.00(0.00) | 1.00 | 604.00(0.00) |
| I don't know/not sure | 8(88.9) | 1(11.1) | 0.99 | 201.60(0.00) | 0.99 | 497.10(0.00) |

**Table legend:** Bivariate (Model I) and multivariate (Model II) logistic regression analysis between demographics, socio-economic characteristics and underweight.

COR = Crude Odds Ratio, AOR = Adjusted Odds Ratio, Sig = Significance level or p-value.

*P < 0•05, 1 = Reference variable in the regression model, OR=Odds Ratio.

as compared to 43.5% of males. This difference was statistically significant, p = 0.01. These findings align with a wide spectrum of global evidence highlighting low physical activity levels among adolescents. For example, the Global School-based Student Health Survey (GSHS) conducted in 23 African countries revealed that adolescent girls are consistently physically inactive than boys, with only 20% of adolescents (25% of males and 16% of females) meeting the WHO-recommended activity levels [23]. Similarly, a pooled analysis of 298 population-based surveys covering 1.6 million adolescents aged 11–17 years from 146 countries identified a global pattern of insufficient physical activity, affecting 81.0% of participants (77.6% of boys and 84.7% of girls), with the highest prevalence reported in high-income Asia Pacific regions [24]. Complementary findings from South Africa, specifically among adolescents from low- and middle-income households, further corroborate this trend [25]. These findings are consistent with those reported in GPAQ-A studies and systematic reviews conducted in Sub-Saharan Africa [26] and Spain [27], reinforcing the global and regional concern regarding adolescent physical inactivity.

This study contrasts with two global surveys. First, findings from 105 countries, including Cambodia, Afghanistan and the Philippines [28] revealed that 57.1% of adolescents engaged in physical activity at least three days per week, a higher trend than observed in our study. Second, a worldwide school-based health survey in low- and middle-income countries [29] reported high physical activity levels in Southeast Asia, particularly in Bangladesh, where adolescents regularly engage in rural labor. These differences highlight regional and contextual variations in adolescent physical activity patterns. Our results corroborate the findings of Muthuri et al [26,30] who also indicated that the overall physical activity levels of teenagers were below the WHO global guidelines [31] for physical activity, which recommend that children and adolescents, 5 – 17 years of age, should engage in a minimum of 60 minutes of daily moderate to vigorous physical activity (MVPA).

This variation in physical activity between males and females could be due to a variation in maturation time between boys and girls, in which boys attain maturation later than girls. Similarly, this difference could also be explained by the fact that in the Kenyan community, boys tend to be more involved in more intense physical work and outdoor activities, such as playing football in fields and playgrounds for longer hours, compared to girls. Because of this, boys tend to have higher energy expenditure when compared to girls. This widening gap in physical activity between boys and girls could also be attributed to a mixture of gender-based stereotypes, body image concerns and existing sociocultural norms that are likely to limit girls' freedom in participating in sports and other physically engaging activities. These aspects can dishearten girls from freely engaging in exercise, particularly in public or competitive settings. Moreover, in urbanized environments such as Mulolongo, the lack of gender-sensitive recreational facilities and worries about safety may further limit access to appropriate fitness facilities for girls, thus reinforcing sedentary lifestyles, restraining opportunities for physical activity.

**Table 8. Crude and adjusted odds ratios from bivariate (Model I) and multivariate (Model II) logistic regression analyses examining associations between demographics, socio-economic characteristics and overweight.**

| Characteristics of the teenager | Overweight | | Model I | | Model II | |
|---|---|---|---|---|---|---|
| | No | Yes | Sig | COR (95%CI) | Sig | AOR (95%CI) |
| | n(%) | n(%) | | | | |
| **Gender** | | | | | | |
| Male | 174(97.8) | 4(2.2) | 1 | | 1 | |
| Female | 149(83.2) | 30(16.8) | 0.00* | 8.75(3.06-25.43) | 0.00* | 8.8(2.98-25.94) |
| **Age cluster of the teenagers** | | | | | | |
| 13-14 years | 68(90.7) | 7(9.3) | 1 | | 1 | |
| 15-19 years | 255(90.4) | 27(9.6) | 0.95 | 1.02(0.43-2.46) | 0.37 | 0.59(0.59-1.85) |
| **Highest level of education** | | | | | | |
| Not finalized | 2(100.0) | 0(0.0) | 1 | | 1 | |
| Primary | 63(95.5) | 3(4.5) | 0.99 | 769.00(0.00) | 0.99 | 314.00(0.00) |
| Secondary | 232(89.6) | 27(10.4) | 0.99 | 188.00(0.00) | 0.99 | 159.00(0.00) |
| University | 19(90.5) | 2(9.5) | 0.99 | 170.00(0.00) | 0.99 | 250.00(0.00) |
| Vocational/technical training | 7(77.8) | 2(22.2) | 0.99 | 461.00(0.00) | 0.99 | 964.00(0.00) |
| **Characteristics of the household** | | | | | | |
| **No. of people in household** | | | | | | |
| 1 to 3 members | 69(94.5) | 4(5.5) | 1 | | 1 | |
| 3 to 6 members | 224(89.2) | 27(10.8) | 0.18 | 2.07(0.70-6.14) | 0.04* | 3.72(2.95-14.63) |
| 7 to 9 members | 30(90.9) | 3(9.1) | 0.49 | 1.72(0.36-8.18) | 0.29 | 2.66(0.42-16.76) |
| **Person responsible for financing the food budget** | | | | | | |
| Father | 180(90.9) | 18(9.1) | 1 | | 1 | |
| Mother | 89(89.0) | 11(11.0) | 0.60 | 1.23(0.56-2.72) | 0.52 | 1.41(0.48-4.13) |
| Grandparents | 3(100.0) | 0(0.0) | 0.99 | 0.00(0.00) | 0.99 | 0.00(0.00) |
| Relatives (uncle, aunt, cousin) | 7(77.8) | 2(22.2) | 0.21 | 2.85(0.55-14.79) | 0.14 | 4.00(0.61-26.23) |
| Self | 5(83.3) | 1(16.7) | 0.53 | 2.00(0.22-18.06) | 0.14 | 7.74(0.49-120.29) |
| Others | 36(94.7) | 2(5.3) | 0.44 | 0.55(0.12-2.50) | 0.46 | 056(0.11-2.68) |
| I don't know/not sure | 3(100.0) | 0(0) | 0.99 | 0.00(0.00) | 0.99 | 0.00(0.00) |
| **Father's highest education level** | | | | | | |
| Not finalized | 1(100.0) | 0(0.0) | | | | |
| Primary | 0(83.3) | 4(16.7) | 1.00 | 323.00(0.00) | 1.00 | 776.00(0.00) |
| Secondary | 73(91.3) | 7(8.8) | 1.00 | 154.00(0.00) | 1.00 | 980.00(0.00) |
| University | 48(87.3) | 7(12.7) | 1.00 | 235.00(0.00) | 1.00 | 150.00(0.00) |
| Vocational/technical training | 27(87.1) | 4(12.9) | 1.00 | 239.00(0.00) | 1.00 | 144.00(0.00) |
| Others | 19(86.4) | 3(13.6) | 1.00 | 255.00(0.00) | 1.00 | 611.00(0.00) |
| I don't know/not sure | 135(93.8) | 9(6.3) | 1.00 | 107.00(0.00) | 1.00 | 538.00(0.00) |
| **Mother's highest education level** | | | | | | |
| Primary | 39(79.6) | 10(20.4) | | | | |
| Secondary | 100(92.6) | 8(7.4) | 0.02* | 0.31(0.11-0.84) | 0.03* | 0.29(0.09-0.93) |
| University | 33(84.6) | 6(15.4) | 0.54 | 0.70(0.23-2.15 | 0.74 | 0.77(0.16-3.60) |
| Vocational/technical training | 28(90.3) | 3(9.7) | 0.21 | 0.41(0.10-1.65) | 0.37 | 0.45(0.08-2.54) |
| Others | 4(100.0) | 0(0.0) | 0.99 | 0.00(0.00) | 0.99 | 0.00(0.00) |
| I don't know/not sure | 119(94.4) | 7(5.6) | 0.00* | 0.22(0.08-0.64) | 0.06 | 0.33(.100-1.09) |
| **The household's main source of income** | | | | | | |
| Agriculture | 3(75.0) | 1(25.0) | | | | |
| Casual labour | 56(86.2) | 9(13.8) | 0.54 | 0.48(0.04-5.15) | 0.63 | 053(0.04-7.14) |

*(Continued)*

**Table 8.** (Continued)

| Characteristics of the teenager | Overweight | | Model I | | Model II | |
|---|---|---|---|---|---|---|
| | No | Yes | Sig | COR (95%CI) | Sig | AOR (95%CI) |
| | n(%) | n(%) | | | | |
| Self-employed running a business | 62(91.2) | 6(8.8) | 0.31 | 0.29(0.02-3.23) | 0.34 | 0.27(0.01-4.00) |
| Salaried employment | 188(92.2) | 16(7.8) | 0.24 | 0.25(0.02-2.59) | 0.25 | 0.22(0.01-2.99) |
| Petty trade | 4(66.7) | 2(33.3) | 0.77 | 1.5(0.08-25.39) | 0.50 | 3.08(0.11-85.28) |
| Others | 1(100.0) | 0(0.0) | 1.00 | 0.00(0.00) | 1.00 | 0.18(0.00) |
| I don't know/not sure | 9(100.0) | 0.0.0) | 0.99 | 0.00(0.00) | 0.99 | 0.00(0.00) |

**Table legend:** Bivariate (Model I) and multivariate (Model II) logistic regression analysis between demographics, socio-economic characteristics and overweight.

COR = Crude Odds Ratio, AOR = Adjusted Odds Ratio, Sig = Significance level or p-value.

*P < 0•05, 1 = Reference variable in the regression model, OR=Odds Ratio.

There is therefore a need to establish gender-supportive infrastructures in Mulolongo, including female-based fitness settings and possibly safe walking pathways, which can encourage girls to indulge more confidently in physical exercises. Community-based strategies and awareness campaigns that can challenge gender stereotyping and promote positive body image can steer a culture of inclusivity and support. Indulging female role models in sporting activities and encouraging peer-led activities can foster participation in physical activity among girls.

These results were also similar to those of the Kenyan Report Card on Physical Activity among children and youth, which showed that Kenyan youths, mainly those residing in urban settings, are gradually becoming physically inactive as compared to those in rural areas [32]. Although our study did not make a comparison between rural and urban settings, Mulolongo ward is within the Nairobi metropolitan region with similar socio-economic characteristics to other urban areas in Kenya. Mulolongo is experiencing rapid urbanization marked by industrial expansion and increasing population density. As a result, available open spaces are progressively being converted into residential homes, thereby limiting the availability of recreational areas such as playgrounds. This constrained spatial environment limits participation in outdoor physical activity among adolescents, who consequently spend more time indoors. Their daily routines tend to revolve around sedentary behaviors like watching television, playing video games and browsing social media, rather than engaging in physically demanding tasks such as walking, fetching water, or participating in outdoor games. This shift in lifestyle patterns is likely to have contributed to reduced levels of physical activity among adolescents in the area. Urban planning should prioritize the preservation and creation of public recreational spaces. The available schools and community centers in Mulolongo can be utilized as dual-purpose facilities to offer structured, safe spaces and comprehensive physical education programs that are equitable across genders. Public health policies should be developed to promote the development of interactive fitness apps, outdoor sporting challenges, such as dancing and movement-based games. These can transform sedentary habits of adolescents into opportunities for physical exertion.

Participation in physical education (PE) classes among teenagers was sporadic, with only 35.9% engaging occasionally. During lunch breaks, a notable (76.2%) of teenagers remained inactive, engaging in sedentary activities such as sitting, talking, reading, or doing schoolwork, with a significant gender disparity (p = 0.04). Physical inactivity persisted after school (51.8%), in the evenings (37.0%), and over weekends (40.6%). Over seven days, (42.3%) of respondents engaged in low physical activity, with females being significantly less active than males (p = 0.00). These findings are similar to those of a study in the South East, South Africa, which demonstrated that teenagers hardly engaged in any physical activity during lunch break, right after school, in the evening and on the weekend [25]. This reaffirms to findings from other studies in Kenya, which reveal that urbanization and academic demands contributed to insufficient physical activity levels among teenagers [8,30]. The low engagement in physical education and general inactivity among the teenagers

in Mulolongo can be linked to numerous interrelated factors. A possible cause for reduced PE participation could be prioritization of academic curricula over physical education. This is compounded by inadequate facilities or a scarcity of trained PE instructors, resulting in inconsistencies in involvement. Furthermore, if PE classes are optional, it can lead to inconsistent participation among students. Beyond normal school hours, it is possible that academic pressures often extend into lunch breaks, with students opting to complete assignments or revise together, thereby contributing to inactivity during these periods. After school, evenings, and weekends, inactivity may stem from teenagers, particularly girls, dedicating time to household chores and caregiving responsibilities. Additionally, with these age groups, there's a growing preference for screentime and browsing over outdoor play, potentially exacerbated by limited access to safe and inclusive recreational infrastructure within the metropolitan setting of Mulolongo. A broader underlying issue could be a general lack of prioritization, engagement, and support from both peers and families regarding participation in sports, games, and exercise.

Together, these findings highlight that well-structured PE classes in schools could play a vital role in helping teenagers achieve the 60 minutes recommended time for PA through physical education. To address low physical activity among teenagers, there is a need to implement policy changes within existing academic curricula to mandate and increase the time allocated for physical education. This includes strict monitoring to ensure consistent student participation in PE classes. Sensitization programs targeting parents and teachers should be implemented to promote adoption and advocacy for healthy, active lifestyles to help establish positive role models within the teenagers' immediate social environment. Community-based programs should be established to provide inclusive, diverse, and accessible physical activity opportunities, ensuring all teenagers can participate in sports, games, and exercise.

This study investigated the nutritional status of teenagers aged 13–19 years old using the BMI for age z-scores [15,13] and the waist circumference cut-offs, as an anthropometric indicator of abdominal obesity, to serve as an independent predictor of insulin resistance and other comorbidities [21,22]. Results revealed that the overall prevalence of severe underweight, moderate underweight, at-risk of overweight, and overweight based on BMI-for-age Z-scores was 1.4%, 6.7%, 9.0%, and 0.3%, respectively. Moreover, there were significant sex variations in the nutritional status of the teenagers; being overweight was more pronounced in females 16.8% while underweight was more pronounced in males 10.1%. Based on waist circumference cut-offs, findings revealed that 3.4% of females were classified as overweight, whereas no males fell into this category. This gender-based disparity was statistically significant, p = 0.01. Additionally, among age groups, 3.6% of females aged 15–19 years were overweight, compared to 2.4% of females aged 13–14 years, a difference that was statistically significant, p = 0.02. Furthermore, the unadjusted bivariate analysis showed that teenagers aged 15–19 years had lower odds of being underweight as compared to those aged between 13–14 years. Some reason contributing to this age specific weight patterns in our study could be that older adolescent girls may experience changes in eating habits and a decrease in physical activity, predisposing them to weight gain. Societal pressures on body dissatisfaction and perceptions of being underweight or overweight can result in both underweight and overweight issues [33].

Some research indicates dissimilar findings where boys generally had higher chance of overweight and obesity compared to girls across adolescent age ranges. For example, girls proved to have a 12% lower likelihood of being obese than boys [34]. Other investigations also reported a higher prevalence of obesity in older boys than in older girls [35]. The prevalence of underweight was comparable to a study done in Addis Ababa, where the prevalence of underweight was 6.2% [36]. The findings from this study are also in line with the nutritional status findings from a literature review in Europe and the Central Asian region, which found that the prevalence of overweight among adolescents was less than 10% in 10 of 11 countries [37]. Although our study found males to be more underweight as compared to females, another study found that adolescent girls in urban slums were underweight and the contributing factors included low parental education, larger family size, and food insecurity [4]. Similarly, thinness among female adolescents in Nepal was associated with pre-mensuration status, fathers' employment conditions, and household food insecurity.

Our findings suggest that females are at a higher risk of abdominal obesity and its associated comorbidities compared to males. These findings align with a study among Indian children and adolescents, where fewer teenagers and specifically females, were less at risk of abdominal obesity based on waist circumference cut-offs when compared with BMI [38]. A similar study conducted in a metropolitan region in India also showed that the prevalence of being overweight and obesity among adolescents was less using abdominal fatness when compared to BMI [39]. This implies that BMI, which gives the indication of weight relative to height for age and sex, may not at all times indicate the adiposity of a person, missing many cases with higher adiposity or misclassifying those with normal abdominal fat percentages as overweight. Equally, abdominal fat percentage in itself might miss certain individuals determined to have higher weight ranges. This study also revealed a significant sex difference between the teenagers, where overweight was more noticeable in females, while undernutrition was more noticeable in males. This finding aligns with the results of the earlier Kenyan Demographic and Health Survey [40] which established that among adolescents aged 15–19, 18% of females were thin, while 43% of males were thin, indicating a higher prevalence of underweight among males, as well as studies conducted in Malawi, Namibia and Zimbabwe [41].

Sex variations in nutritional status have been associated with complex interactions between the social, cultural, biological and environmental aspects [42]. Literature review shows that the risk of overweight and obesity in female teenagers might be attributed to the start of menstruation, its related hormonal fluctuations and psychological effect on physical activity [5]. On the contrary, male teenagers at a similar age mark a rise in energy needs associated with increased engagement in physical activities, resulting in reduced energy storage. The urbanized setting of Mulolongo likely favors an obesogenic environment characterized by increased exposure to fast food outlets. Such environments influence shifts in dietary patterns toward energy-dense, nutrient-poor foods, while at the same time limiting opportunities for physical activity due to infrastructural constraints or lifestyle transitions. Girls, in particular, may experience more sedentary lifestyles due to societal expectations around domestic roles, safety concerns, or limited access to recreational spaces. Furthermore, households within Mulolongo likely depend on a market-driven food system characterized by greater access to food availability and access that may lead to inequities in dietary intake. These aspects, together with urbanized dietary shifts, favor the intake of energy-dense, processed foods and sedentary lifestyles may help explain the increased risk of overweight and obesity among adolescents in such settings. Tackling underweight in teenagers is vital as this stage is a critical period for physical growth, cognitive development, and emotional well-being [43]. Failure to address undernutrition can lead to stunted growth, weakened immunity, delayed puberty, and increased risk of osteoporosis and chronic diseases later in life [3]. Besides, being overweight and obese in childhood and adolescence increases the risk of cardiovascular diseases, type 2 diabetes, hypertension, and metabolic syndrome in adulthood [44].

Interestingly, teenagers from households with 3–6 members had distinctly higher odds of being overweight as compared to those from smaller households with 1–3 members. This is a very unique observation, contrary to many previous studies. For example, a study among Malaysian adolescents [45] showed an inverse relationship between household size and BMI z-scores, indicating that larger households were associated with lower BMI. Similarly, another study [46] reported that each additional family member reduced the likelihood of weight gain in children, suggesting a protective effect of bigger family size, opposing to the trend observed in our study. However, when comparing our findings to other studies, it is important to note that family structure, such as living with biological parents versus other caregivers, may have a substantial influence on BMI than simply the number of people in a household. This suggests that factors beyond household size play a role in adolescent weight status. These factors could include the person responsible for financing the food budget and the source of income, among others, even though these were not statistically significant in our model. Further research is needed to identify contributing factors.

Maternal education was associated with higher odds of being overweight in our study. Similarly, maternal education levels were associated with increased child overweight and obesity in a study conducted among children and adolescents in Saudi Arabia [47] and another among teenagers in Tibetan [48]. Another research in Nigeria reported a 10.2% prevalence

of overweight/obesity among female adolescents, with higher household wealth being a significant determinant [49]. Much evidence on the association between adolescent parents' education and overweight or obesity varies [50]. Whereas some show an attenuated risk [51] the opposite was reported in others [52]. In our study, respondents who reported that their mothers had secondary education had higher odds of being overweight as compared to those whose mothers had primary education. There are two possible explanations for these opposing phenomena. First, the positive association might occur because educated mothers are more likely to have higher incomes, which are linked to greater access to energy-dense foods and electronic devices that promote sedentary behavior, thus increasing the risk of overweight or obesity. On the other hand, higher maternal education could lead to better knowledge of good nutritional practices, which might explain our findings. However, this relationship could be influenced by other factors, so further studies are needed to clarify it.

The study provides valuable insights, though some limitations should be acknowledged. First, its metropolitan focus (Mulolongo Ward) limits the generalizability of findings to rural Kenyan teens, who may typically engage in more agrarian labor [8]. Second, self-reported PA data (via GPAQ-A), though a standardized and validated tool, may tend to overestimate activity due to recall bias [53]. A longitudinal analysis incorporating real-time physical activity monitoring may yield a more robust measure of physical behaviours. Third, the cross-sectional nature of the study precludes drawing causal inferences, for instance, whether maternal education directly increases obesity risk or reflects broader socioeconomic factors, which proved not to be statistically significant. Further investigation should thus be done to identify and correlate the potential determinants of nutritional status among adolescents. Finally, even though BMI is still an acceptable tool for measuring nutritional status, its inability to distinguish fat from muscle mass may misclassify healthy, muscular teens as overweight [54] while waist circumference cut-offs may lack sensitivity for adolescents [22]. These limitations suggest that the true prevalence of inactivity and obesity could be either higher or differently distributed. Findings from this study are most applicable to Kenyan youths in other metropolitan regions with similar socio-economic settings. However, gender disparities in PA and nutrition outcomes resonate across LMICs, where girls often face cultural barriers to exercise and tend to have higher obesity risks post-puberty [55]. In contrast, high-income countries report higher female PA participation, suggesting that infrastructure and gender norms play pivotal roles [56]. The combined use of BMI-for-age Z-scores, waist circumference cut-off points and behavioral (GPAQ activity scores) metrics provides compelling evidence in the assessment of nutritional status for this study. Future studies should incorporate a combination of indicators when assessing nutritional status, such as biochemical and dietary indicators.

## Conclusion

This cross-sectional study was conducted to assess the relationship between physical activity levels and nutritional status among adolescents in Machakos County, Kenya. Through questionnaires and anthropometric measurements, the study found that many teenagers were physically inactive, with differences observed between genders. Regarding nutritional status, the study identified instances of mild underweight and overweight, with variations between male and female adolescents. The findings highlight the link between physical inactivity and poor nutritional outcomes, suggesting the need for targeted public health interventions to improve physical activity levels and nutritional status among adolescents, thereby promoting their overall health. The existing school-based curriculum should be revised to include simple and targeted nutrition education sessions to promote healthy food choices among students. Strict monitoring should be done to ensure consistent student participation in PE classes. Sensitization programs targeting parents and families in households, such as nutrition talks by healthcare workers, should be implemented to promote adoption and advocacy for healthy, active lifestyles to help establish positive role models within the teenagers' immediate social environment. In summary, this study provides a robust empirical foundation for planning comparative and longitudinal or interventional studies to better assess causal relationships between rural and urban settings and help identify contextual influences on health behaviors.

## Supporting information

**S1 Table. Structured questionnaire for teenagers.**
(DOCX)

**S1 Dataset. Minimal dataset.**
(XLSX)

## Acknowledgments

The authors express their gratitude to all the teenagers who participated in this study, the parents who gave permission for their children to take part and the relevant authorities for allowing the study to be conducted.

## Author contributions

**Conceptualization:** Moses Amram Kutwah.

**Data curation:** Moses Amram Kutwah.

**Formal analysis:** Moses Amram Kutwah.

**Funding acquisition:** Moses Amram Kutwah.

**Investigation:** Moses Amram Kutwah.

**Methodology:** Moses Amram Kutwah.

**Project administration:** Moses Amram Kutwah.

**Resources:** Moses Amram Kutwah.

**Software:** Moses Amram Kutwah.

**Supervision:** Dorcus Mbithe David-Kigaru, Joseph Kobia.

**Validation:** Moses Amram Kutwah.

**Visualization:** Moses Amram Kutwah.

**Writing – original draft:** Moses Amram Kutwah, Dorcus Mbithe David-Kigaru.

**Writing – review & editing:** Moses Amram Kutwah, Dorcus Mbithe David-Kigaru.

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
