## [Decision Letter · Decision Letter 0]

25 Sep 2025

PGPH-D-25-01337

Are Teenagers in Kenya physically active? The nexus between physical activity and Nutrition status of Kenyan Teenagers: A cross sectional study

Dear Dr. Kutwah,

Thank you for submitting your manuscript to PLOS Global Public Health. After careful consideration, we feel that it has merit but does not fully meet PLOS Global Public Health’s publication criteria as it currently stands. Therefore, we invite you to submit a revised version of the manuscript that addresses the points raised during the review process.

We look forward to receiving your revised manuscript.

Kind regards,

Madhur Verma

Academic Editor

Journal Requirements:

Additional Editor Comments (if provided):

Reviewer #1:

Reviewer #2:

Reviewers' comments:

Reviewer's Responses to Questions

**Comments to the Author**

1. Does this manuscript meet PLOS Global Public Health’s publication criteria?

Reviewer #1: Yes

Reviewer #2: Yes

2. Has the statistical analysis been performed appropriately and rigorously?

Reviewer #1: Yes

Reviewer #2: Yes

3. Have the authors made all data underlying the findings in their manuscript fully available (please refer to the Data Availability Statement at the start of the manuscript PDF file)?

Reviewer #1: No

Reviewer #2: No

4. Is the manuscript presented in an intelligible fashion and written in standard English?

Reviewer #1: No

Reviewer #2: Yes

Reviewer #1: A few questions remained unclear to me. 1. Was there permission from the ethics committee for the study? If so, you need the name of the ethics committee and the decision number. 2. Has written voluntary informed consent been obtained to participate in the study? If so, did your parents/guardians give it to you? Persons under the age of 16 (at least) cannot express such consent themselves without the participation of a legal representative. 3. How were the principles of non-disclosure/confidentiality of information observed?

The authors need to clarify/supplement the following points. 1. Give a detailed description of the nutrition questionnaire. What was analyzed? 2. Provide ethnic characteristics in the results. 3. In the discussion, it is necessary to compare our own data with the world literature. In which countries is the situation similar? Where is the situation different? 4. In conclusion, I must say that work provides benefits for public health. It is possible to discuss preventive measures in the discussion.

Reviewer #2: This cross-sectional study was conducted to assess the relationship between physical activity levels and nutritional status among adolescents in Machakos County, Kenya. Through questionnaires and anthropometric measurements, the study found that many adolescents were physically inactive, with differences observed between genders. Regarding nutritional status, the study identified instances of mild underweight and overweight, with variations between male and female adolescents. The findings highlight the link between physical inactivity and poor nutritional outcomes, suggesting the need for targeted public health interventions to improve physical activity levels and nutritional status among adolescents, thereby promoting their overall health. I have some minor comments for future improvement.

1. Although specific methods of data collection are introduced, the details are not sufficient, such as the content of the questionnaire and how to ensure its reliability and validity.

2. The authors should add a paragraph explaining the choice of statistical analysis methods, clarifying why this method is suitable for the data and research questions of this study. Also, describe in detail how confounding factors were controlled to improve the accuracy and reliability of the analysis results.

3. Although the consistency and differences between the study results and existing research are discussed, the in-depth explanation of the results is still insufficient, lacking exploration of possible mechanisms and influencing factors.

4. Some limitations are mentioned in the discussion section, but the impact of these limitations on the study results is not fully discussed, nor are suggestions for improvement in future studies provided.

5. In the Conclusion, the authors should provide more specific suggestions, including detailed steps for implementing these interventions and potential challenges.

**Do you want your identity to be public for this peer review?** For information about this choice, including consent withdrawal, please see our Privacy Policy

Reviewer #1: No

Reviewer #2: **Yes:** Pengpeng Ye

---

## [Decision Letter · Decision Letter 1]

29 Oct 2025

PGPH-D-25-01337R1

Are Teenagers in Kenya Physically Active? The Nexus Between Physical Activity and Nutrition Status of Kenyan Teenagers: A Cross Sectional Study

Dear Dr. Kutwah,

Thank you for submitting your manuscript to PLOS Global Public Health. After careful consideration, we feel that it has merit but does not fully meet PLOS Global Public Health’s publication criteria as it currently stands. Therefore, we invite you to submit a revised version of the manuscript that addresses the points raised during the review process.

We look forward to receiving your revised manuscript.

Kind regards,

Madhur Verma

Academic Editor

Journal Requirements:

Reviewers' comments:

Reviewer's Responses to Questions

**Comments to the Author**

Reviewer #1: All comments have been addressed

Reviewer #2: All comments have been addressed

publication criteria?

Reviewer #1: Yes

Reviewer #2: Yes

3. Has the statistical analysis been performed appropriately and rigorously?

Reviewer #1: Yes

Reviewer #2: Yes

4. Have the authors made all data underlying the findings in their manuscript fully available (please refer to the Data Availability Statement at the start of the manuscript PDF file)?

Reviewer #1: Yes

Reviewer #2: Yes

5. Is the manuscript presented in an intelligible fashion and written in standard English?

Reviewer #1: Yes

Reviewer #2: Yes

Reviewer #1: Authors took into account all reviewer's comments

Reviewer #2: I have no additional comments.

**Do you want your identity to be public for this peer review?** For information about this choice, including consent withdrawal, please see our Privacy Policy

Reviewer #1: No

Reviewer #2: **Yes:** Pengpeng Ye

---

## [Editor Report · Decision Letter 2]

23 Dec 2025

Are Teenagers in Kenya Physically Active? The Nexus Between Physical Activity and Nutrition Status of Kenyan Teenagers: A Cross Sectional Study

PGPH-D-25-01337R2

Dear Mr. Kutwah,

We are pleased to inform you that your manuscript 'Are Teenagers in Kenya Physically Active? The Nexus Between Physical Activity and Nutrition Status of Kenyan Teenagers: A Cross Sectional Study' has been provisionally accepted for publication in PLOS Global Public Health.

Best regards,

Madhur Verma

Academic Editor